# Exciton diffusion in two-dimensional metal-halide perovskites

Michael Seitz [1,2], Alvaro J. Magdaleno[1,2], Nerea Alcázar-Cano [1,3], Marc Meléndez [1,3], Tim J. Lubbers[1,2], Sanne W. Walraven [1,2], Sahar Pakdel [4], Elsa Prada [1,2], Rafael Delgado-Buscalioni [1,3] & Ferry Prins [1,2✉]

Two-dimensional layered perovskites are attracting increasing attention as more robust analogues to the conventional three-dimensional metal-halide perovskites for both light harvesting and light emitting applications. However, the impact of the reduced dimensionality on the optoelectronic properties remains unclear, particularly regarding the spatial dynamics of the excitonic excited state within the two-dimensional plane. Here, we present direct measurements of exciton transport in single-crystalline layered perovskites. Using transient photoluminescence microscopy, we show that excitons undergo an initial fast diffusion through the crystalline plane, followed by a slower subdiffusive regime as excitons get trapped. Interestingly, the early intrinsic diffusivity depends sensitively on the choice of organic spacer. A clear correlation between lattice stiffness and diffusivity is found, suggesting exciton–phonon interactions to be dominant in the spatial dynamics of the excitons in perovskites, consistent with the formation of exciton–polarons. Our findings provide a clear design strategy to optimize exciton transport in these systems.

[1] Condensed Matter Physics Center (IFIMAC), Autonomous University of Madrid, 28049 Madrid, Spain. [2] Department of Condensed Matter Physics, Autonomous University of Madrid, 28049 Madrid, Spain. [3] Department of Theoretical Condensed Matter Physics, Autonomous University of Madrid, 28049 Madrid, Spain. [4] Department of Physics and Astronomy, Aarhus University, 8000 Aarhus C, Denmark. ✉email: ferry.prins@uam.es

Metal-halide perovskites are a versatile material platform for light harvesting[1–4] and light emitting applications[5,6], combining the advantages of solution processability with high ambipolar charge carrier mobilities[7,8], high defect tolerance[9–11], and tunable optical properties[12,13]. Currently, the main challenge in the applicability of perovskites is their poor environmental stability[14–17]. Reducing the dimensionality of perovskites has proven to be one of the most promising strategies to yield a more stable performance[17–19]. Perovskite solar cells with mixed two-dimensional (2D) and three-dimensional (3D) phases, for example, have been fabricated with efficiencies above 22%[20] and stable performance for more than 10,000 h[21], while phase pure 2D perovskite solar cells have been reported with efficiencies above 18%[22,23]. Likewise, significant stability improvements have been reported for phase pure 2D perovskites as the active layer in light emitting technologies[24–29]. The improved environmental stability in 2D perovskite phases is attributed to a better moisture resistance due to the hydrophobic organic spacers that passivate the inorganic perovskite sheets, as well as an increased formation energy of the material[17–19,30].

However, the reduced dimensionality of 2D perovskites dramatically affects the charge carrier dynamics in the material, requiring careful consideration in their application in optoelectronic devices[31–33]. 2D perovskites are composed of inorganic metal-halide layers, which are separated by long organic spacer molecules. They are described by their general chemical formula $L_2[ABX_3]_{n-1}BX_4$, where A is a small cation (e.g. methylammonium, formamidinium), B is a divalent metal cation (e.g. lead, tin), X is a halide anion (chloride, bromide, iodide), L is a long organic spacer molecule, and n is the number of octahedra that make up the thickness of the inorganic layer. The separation into few-atom thick inorganic layers yields strong quantum and dielectric confinement effects[34]. As a result, the exciton binding energies in 2D perovskites can be as high as several hundreds of meVs, which is around an order of magnitude larger than those found in bulk perovskites[35–37]. The excitonic character of the excited state is accompanied by an effective widening of the bandgap, an increase in the oscillator strength, and a narrowing of the emission spectrum[36–38]. The strongest confinement effects are observed for $n = 1$, where the excited state is confined to a single B-X-octahedral layer.

Consequently, light harvesting using 2D perovskites relies on the efficient transport of excitons and their subsequent separation into free charges[39]. This stands in contrast to bulk perovskites in which free charges are generated instantaneously thanks to the small exciton binding energy[35]. Particularly, with excitons being neutral quasi-particles, the charge extraction becomes significantly more challenging as they cannot be guided to the electrodes through an external electric field[40]. Excitons need to diffuse to an interface before the electron and hole can be efficiently separated into free charges[41]. On the other hand, for light emitting applications the spatial displacement is preferably inhibited, as a larger diffusion path increases the risk of encountering quenching sites which would reduce brightness. While charge transport in bulk perovskites has been studied in great detail, the mechanisms that dictate exciton transport in 2D perovskites remain elusive[41]. Moreover, it is unclear to what extent exciton transport is influenced by variations in the perovskite composition.

Here, we report the direct visualization of exciton diffusion in 2D single-crystalline perovskites using transient photoluminescence-microscopy[42]. This technique allows us to follow the temporal evolution of a near-diffraction-limited exciton population with sub-nanosecond resolution and reveals the spatial and temporal exciton dynamics. We observe two different diffusion regimes. For early times, excitons follow normal diffusion, while for later times a subdiffusive regime emerges, which is attributed to the presence of trap states. Using the versatility of perovskite materials, we study the influence of the organic spacer on the diffusion dynamics of excitons in 2D perovskites. We find that between commonly used organic spacers (phenethylammonium, PEA, and butylammonium, BA), diffusivities and diffusion lengths can differ by one order of magnitude. We show that these changes are closely correlated with variations in the softness of the lattice, suggesting a dominant role for exciton–phonon coupling and exciton–polaron formation in the spatial dynamics of excitons in these materials. These insights provide a clear design strategy to further improve the performance of 2D perovskite solar cells and light emitting devices.

## Results

**Exciton diffusion imaging.** We prepare single crystals of $n = 1$ phenethylammonium lead iodine $(PEA)_2PbI_4$ 2D perovskite by drop-casting a saturated precursor solution onto a glass substrate[43,44], as confirmed by XRD analysis and photoluminescence spectroscopy (see Methods section for details). Using mechanical exfoliation, we isolate single-crystalline flakes of the perovskite and transfer these to microscopy slides. The single-crystalline flakes have typical lateral sizes of tens to hundreds of micrometers and are optically thick. The use of thick flakes provides a form of self-passivation that prevents the typical fast degradation of the perovskite in ambient conditions.

To measure the temporal and spatial exciton dynamics, we create a near-diffraction-limited exciton population using a pulsed laser diode ($\lambda_{ex} = 405$ nm) and an oil immersion objective (N.A. = 1.3). The image of the fluorescence emission of the exciton population is projected outside the microscope with high magnification (×330), as illustrated in Fig. 1b. By placing a scanning avalanche photodiode (20 μm in size) in the image plane, we resolve the time-dependent broadening of the population with high temporal and spatial resolution. Fig. 1c shows the resulting map of the evolution in space and time of the fluorescence emission intensity of an exciton population in $(PEA)_2PbI_4$. The fluorescence emission intensity $I(x,t)$ is normalized at each point in time to highlight the broadening of the emission spot over time. Each time-slice $I(x,t_c)$ is well described by a Voigt function[45], from which we can extract the variance $\sigma(t)^2$ of the exciton distribution at each point in time (Fig. 1d). On a timescale of several nanoseconds, the exciton distribution broadens from an initial $\sigma(t = 0 \text{ ns}) = 171$ nm to $\sigma(t = 10 \text{ ns}) = 448$ nm, indicating fast exciton diffusion.

To analyze the time-dependent broadening of the emission spot in more detail, we study the temporal evolution of the mean-square-displacement (MSD) of the exciton population, given by $MSD(t) = \sigma(t)^2 - \sigma(0)^2$. Taking the one-dimensional diffusion equation as a simple approximation, it follows that $MSD(t) = 2Dt^\alpha$, which allows us to extract the diffusivity $D$ and the diffusion exponent $\alpha$ from our measurement (see Supplementary Note 1)[42,45]. In Fig. 1e we plot the MSD as a function of time. Two distinct regimes can be observed: For early times ($t \lesssim 1$ ns) a fast linear broadening occurs with $\alpha = 1.01 \pm 0.01$, indicative of normal diffusion, while for later times ($t \gtrsim 1$ ns) the broadening becomes progressively slower with $\alpha = 0.65 \pm 0.01$, suggesting a regime of trap state limited exciton transport (see Supplementary Note 2). The two regimes are clearly visible in the log–log representation shown in the inset of Fig. 1e, where different slopes correspond to different $\alpha$ values. From these measurements, a diffusivity of $0.192 \pm 0.013$ cm$^2$ s$^{-1}$ is found for $(PEA)_2PbI_4$. Our diffusivity of single crystalline $(PEA)_2PbI_4$ is around an order of magnitude higher than

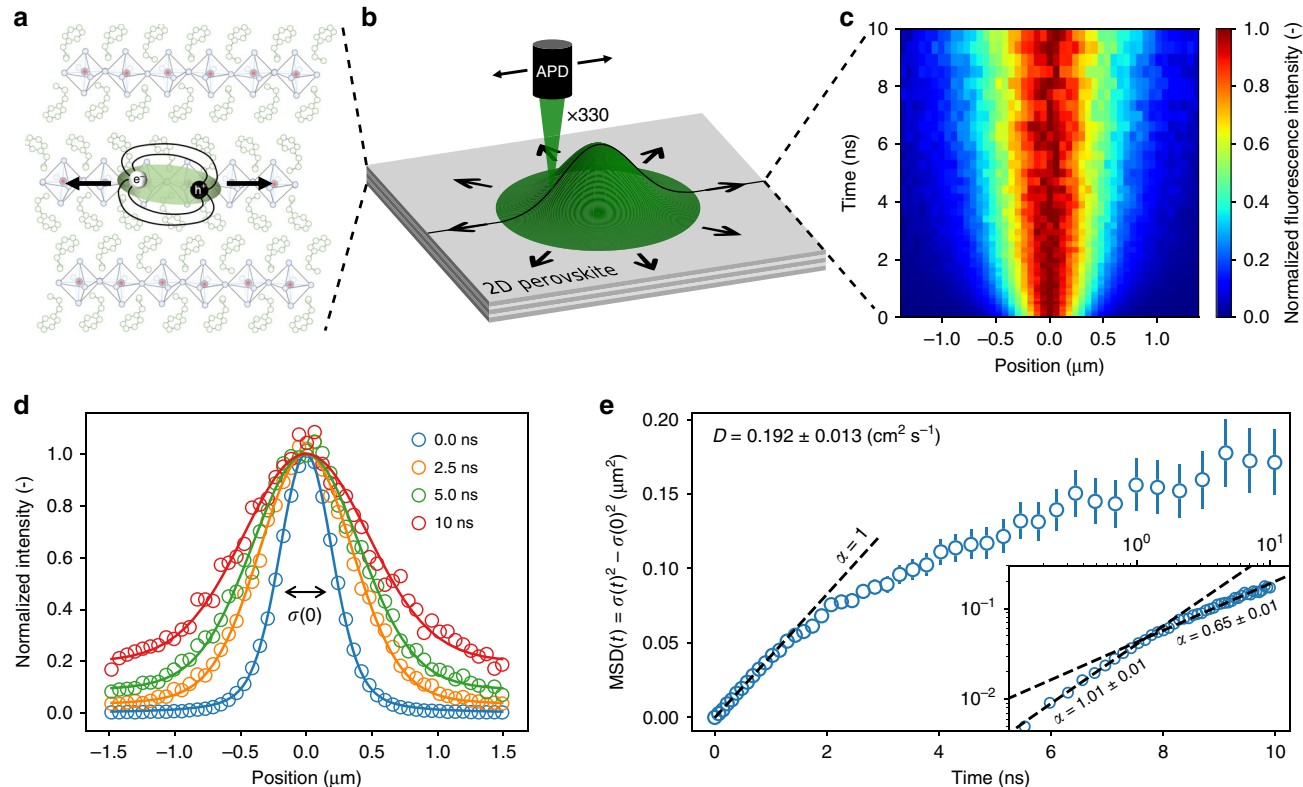

**Fig. 1 Diffusion imaging of excitons in two-dimensional perovskites. a** Illustration of the $(PEA)_2PbI_4$ crystal structure, showing the perovskite octahedra sandwiched between the organic spacer molecules. **b** Schematic of the experimental setup. A near-diffraction limited exciton population is generated with a pulsed laser diode. The spatial and temporal evolution of the exciton population is recorded by scanning an avalanche photodiode through the magnified image of the fluorescence I(x,t). **c** Fluorescence emission intensity I(x,t) normalized at each point in time to highlight the spreading of the excitons. **d** Cross section of I(x,t) for different times $t_c$. **e** Mean-square-displacement of the exciton population over time. Two distinct regimes are present: First, normal diffusion with $\alpha = 1$ is observed, which is followed by a subdiffusive regime with $\alpha < 1$. The inset shows a log–log plot of the same data, highlighting the two distinct regimes. Reported errors represent the uncertainty in the fitting procedure for $\sigma(t)^2$.

previously reported mobility values from conductivity measurements ($\mu = 1\,cm^2\,V^{-1}\,s^{-1}$; $D = \mu k_B T = 0.025\,cm^2\,s^{-1}$) of polycrystalline films[46]. This finding is reasonable as grain boundaries slow down the movement of excitons and conventional methods measure a time-averaged mobility that cannot separate intrinsic diffusion from trap state limited diffusion.

**Influence of trap states**. The role of trap states in perovskite materials is well studied and is generally attributed to the presence of imperfections at the surface of the inorganic layer[47]. These lower-energy sites lead to a subdiffusive behavior as a subpopulation of excitons becomes trapped. To test the influence of trap states, we have performed diffusion measurements in the presence of a continuous wave (CW) background excitation of varying intensity (Fig. 2). The background excitation leads to a steady state population of excitons, which fill some of the traps and thereby reduce the effective trap density. To minimize the invasiveness of the measurement itself, the repetition rate and fluence were reduced to a minimum (see Supplementary Note 2). In the absence of any background illumination, we find a strongly subdiffusive diffusion exponent of $\alpha = 0.48 \pm 0.02$. As the background intensity is increased, an increasing $\alpha$ is observed, indicative of trap state filling. Ultimately, a complete elimination of subdiffusion ($\alpha = 0.99 \pm 0.02$) is obtained at a background illumination power of 60 mW cm$^{-2}$. For comparison, this value corresponds roughly to a 2.5 Sun illumination. Additionally, we observe that the onset of the subdiffusive regime is delayed as

more and more trap states are filled, as represented by the increasing $t_{split}$ parameter (see Fig. 2b, bottom panel).

To gain theoretical insights and quantitative predictions concerning the observed subdiffusive behavior of excitons and its relation with trap state densities, we performed numerical simulations based on Brownian dynamics of individual excitons diffusing in a homogeneously distributed and random trap field (see Supplementary Note 3). In addition, we developed a coarse-grained theoretical model based on continuum diffusion of the exciton concentration (see Supplementary Note 4). The continuum theory predicts an exponential decay of the diffusion coefficient,

$$\frac{1}{2}\frac{dMSD(t)}{dt} = D(t) = D\exp\left(-\frac{D}{\lambda^2}t\right) \qquad (1)$$

where $\lambda$ is the average distance between traps. The integral of this expression leads to

$$MSD(t) = 2\lambda^2\left[1 - \exp\left(-\frac{D}{\lambda^2}t\right)\right], \qquad (2)$$

which, as shown in Fig. 2c, successfully reproduces both experimental and numerical results and allows us to determine the value of the intrinsic trap state density, yielding $1/\lambda^2 = 22\,\mu m^{-2}$ per layer ($\approx 10^{16}\,cm^{-3}$), which is of the same order of magnitude as previously reported values for bulk perovskites[48,49]. The inset in Fig. 2c shows the evolution of the effective trap state density $1/\lambda^2$ with increasing illumination intensity. We note that the exponential decay of Eq. 1 allows for a more intuitive characterization of D(t) by

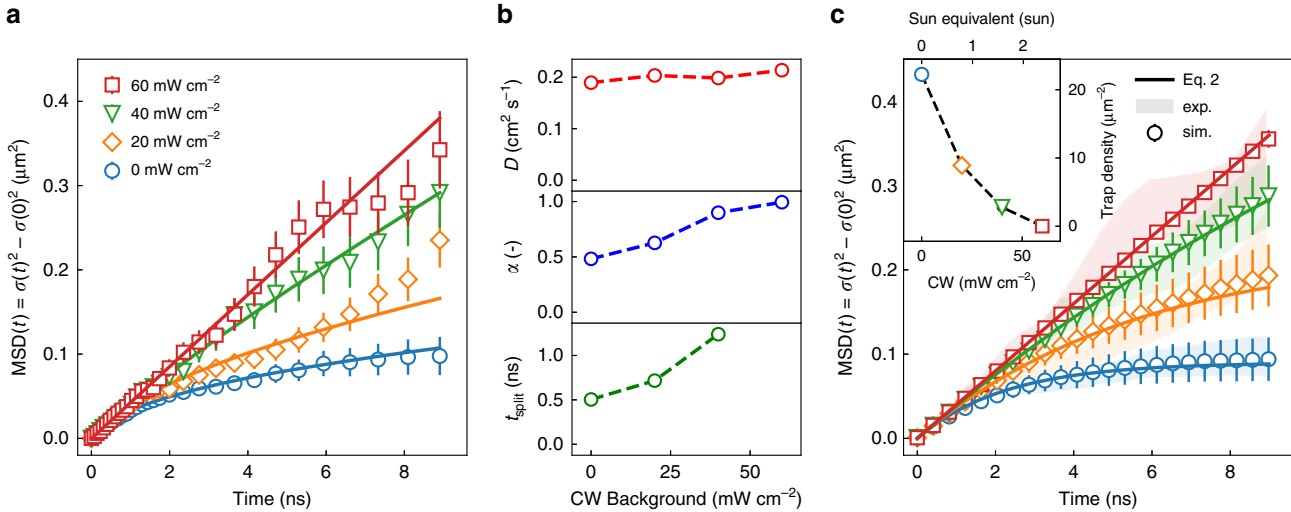

**Fig. 2 Exciton diffusion with different background excitation intensities. a** Mean-square-displacement of the exciton population for different continuous wave (CW) background intensities. Experimental values are displayed with open markers, while the fit functions (Supplementary Eq. 9), defined through the parameters $D$, $\alpha$, and $t_{split}$, are displayed as solid lines. Reported errors represent the uncertainty in the fitting procedure for $\sigma(t)^2$. **b** Diffusivity $D$, diffusion exponent $\alpha$, and the onset of subdiffusive regime $t_{split}$ extracted from fits in **a**. **c** Theoretical model (Eq. 2, solid lines), and numerical simulation (open markers) for exciton diffusion with different trap densities. Experimental values from **a** are displayed as shaded areas for comparison. Reported errors represent the standard deviation of $10^4$ Brownian motion simulations. The inset shows the trap densities found with the simulations. Mirror axis of the inset is the sun equivalent of the background illumination intensity (AM1.5 Global with $E_{photon} > E_{bandgap}$).

relating the subdiffusion directly to the trap density $1/\lambda^2$ rather than relying on the subdiffusive exponent $\alpha$ of a power law commonly used in literature[42].

**Structure-property relations of exciton transport.** Importantly, the early diffusion dynamics is unaffected by the trap density and shows normal diffusion ($\alpha = 1$) for all illumination intensities. This strongly suggests that the early diffusion dynamics is unaffected by energetic disorder, which would result in a sublinear behavior with $\alpha < 1$, and any trap states, giving us direct access to the intrinsic exciton diffusivity of the material and allowing us to compare the intrinsic exciton diffusivity between perovskites of different compositions. To explore compositional variations, we substitute phenethylammonium (PEA) with butylammonium (BA)—another commonly used spacer molecule for 2D perovskites[18,24,25,28,39,50,51].

Fig. 3a displays the MSD of the $(BA)_2PbI_4$ perovskite, again showing the distinct transition from normal diffusion to a subdiffusive regime. However, as compared to $(PEA)_2PbI_4$, excitons in $(BA)_2PbI_4$ are remarkably less mobile, displaying a diffusivity of only $0.013 \pm 0.002$ $cm^2$ $s^{-1}$, which is over an order of magnitude smaller than that of $(PEA)_2PbI_4$ with $0.192 \pm 0.013$ $cm^2$ $s^{-1}$ (green curve shown in Fig. 3b for comparison). Taking the exciton lifetime into account, the difference in diffusivity results in a reduction in the diffusion length from $236 \pm 4$ nm for $(PEA)_2PbI_4$ to a mere $39 \pm 8$ nm for $(BA)_2PbI_4$ (see Fig. 3c and Supplementary Note 5). These results indicate that the choice of ligand plays a crucial role in controlling the spatial dynamics of excitons in 2D perovskites. We would like to note that the reported diffusion lengths follow the literature convention of diffusion lengths in one dimension, as it is the relevant length scale for device design. The actual 2D diffusion length is greater by a factor of $\sqrt{2}$.

To understand the large difference in diffusivity between $(PEA)_2PbI_4$ and $(BA)_2PbI_4$, we take a closer look at the structural differences between these two materials. Changing the organic spacer can have a significant influence on the structural and optoelectronic properties of 2D perovskites. Specifically,

increasing the cross-sectional area of the organic spacer distorts the inorganic lattice and reduces the orbital overlap between neighboring octahedra, which in turn increases the effective mass of the exciton[52]. Comparing the octahedral tilt angles of $(PEA)_2PbI_4$ and $(BA)_2PbI_4$, a larger distortion for the bulkier $(PEA)_2PbI_4$ (152.8°) as compared to $(BA)_2PbI_4$ (155.1°) is found[53,54]. The larger exciton effective mass in $(PEA)_2PbI_4$ would, however, suggest slower diffusion, meaning a simple effective mass picture for free excitons cannot explain the observed trend in the diffusivity between $(PEA)_2PbI_4$ and $(BA)_2PbI_4$.

Recently, exciton–phonon interactions have been found to strongly influence exciton dynamics in perovskites[31,33]. To investigate the possible role of exciton–phonon coupling on exciton diffusion, we first quantify the softness of the lattices of both $(PEA)_2PbI_4$ and $(BA)_2PbI_4$ by extracting the atomic displacement parameters from their respective single crystal X-ray data[55]. The atomic displacement of the different atoms of both systems are summarized in Fig. 3d, showing distinctly larger displacements for $(BA)_2PbI_4$ as compared to $(PEA)_2PbI_4$ in both the organic and inorganic sublattice[53,54]. The increased lattice rigidity for $(PEA)_2PbI_4$ can be attributed to the formation of an extensive network of pi-hydrogen bonds and a more space-filling nature of the aromatic ring, both of which are absent in the aliphatic BA spacer molecule. Qualitatively, a stiffening of the lattice reduces the exciton–phonon coupling and would explain the observed higher diffusivity in $(PEA)_2PbI_4$ as compared to $(BA)_2PbI_4$. In addition to a softer lattice, we find that $(BA)_2PbI_4$ exhibits a larger exciton–phonon coupling strength than $(PEA)_2PbI_4$, as confirmed by analyzing the temperature-dependent broadening of the photoluminescence linewidth of the two materials (see Supplementary Note 6)[56].

To further test the correlation between lattice softness and diffusivity, we have performed measurements on a wider range of 2D perovskites with different organic spacers. In Fig. 3e, we present the diffusivity as a function of average atomic displacement for each of the different perovskite unit cells. Across the entire range of organic spacers, a clear correlation between the diffusivity and the lattice softness is found, further confirming the dominant role of

**Fig. 3 Exciton diffusion in (PEA)$_2$PbI$_4$ and (BA)$_2$PbI$_4$.** **a** (PEA)$_2$PbI$_4$ and (BA)$_2$PbI$_4$ crystal structure along the a crystal axis[53,54]. **b** Mean-square-displacement of exciton population over time for (PEA)$_2$PbI$_4$ (dotted line) and (BA)$_2$PbI$_4$ (circles). Inset shows the normalized fluorescence emission intensity I($x$,$t$) for (BA)$_2$PbI$_4$. Reported errors represent the uncertainty in the fitting procedure for $\sigma(t)^2$. **c** Fractions of surviving excitons (extracted from lifetime data in Supplementary Fig. 7) as a function of net spatial displacement $\sqrt{\text{MSD}(t)}$ of excitons for (PEA)$_2$PbI$_4$ (triangles) and (BA)$_2$PbI$_4$ (circles). Reported errors represent the uncertainty in the fitting procedure for $\sigma(t)^2$. **d** Average atomic displacement $U_{eq}$ of the chemical elements in (PEA)$_2$PbI$_4$ and (BA)$_2$PbI$_4$. Data was extracted from previously published single crystal X-ray diffraction data[53,54]. **e** Diffusivity $D$ as a function of average atomic displacement $U_{eq}$ for different organic spacers: 4-fluoro-phenethylammonium (4FPEA)[65] phenethylammonium (PEA)[53], hexylammonium (HA)[54], octylammonium (OA)[66], decylammonium (DA)[66], Butylammonium (BA)[54]. Reported errors represent the standard deviation of the average diffusivity $D$ obtained from multiple single crystalline flakes.

exciton–phonon coupling in the spatial dynamics of the excited state in 2D perovskites.

In the limit of strong exciton–phonon coupling, the presence of an exciton could potentially cause distortions of the soft inorganic lattice of the perovskite and lead to the formation of exciton–polarons[57,58]. As compared to a free exciton, an exciton–polaron would exhibit a larger effective mass and, consequently, a lower diffusivity. The softer the lattice, the larger the distortion, and the heavier the polaron effective mass would be[59].

Polaron formation can significantly modify the mechanism of transport, in some cases causing a transition from band-like to a hopping type transport[59]. When short-range deformations of the lattice are dominant, the exciton–polaron is localized within a unit cell of the material and is known as a small polaron. The motion of small polarons occurs through site-to-site hopping and increases with temperature ($\partial D/\partial T > 0$). However, in the presence of dominant long-range lattice deformations, large exciton–polarons may form which extend across multiple lattice sites. The diffusion of large polarons decreases with increasing temperature ($\partial D/\partial T < 0$), resembling that of band-like free exciton motion, although with

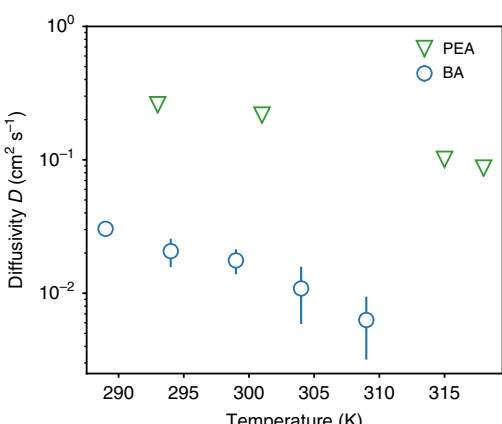

**Fig. 4 Temperature-dependent diffusivity in (PEA)$_2$PbI$_4$ (triangles) and (BA)$_2$PbI$_4$ (circles).** Error bars represent the uncertainty of the fit and are smaller than the markers for (PEA)$_2$PbI$_4$.

a strongly increased effective mass. In Fig. 4, we present temperature-dependent measurements of the diffusivity for both $(PEA)_2PbI_4$ and $(BA)_2PbI_4$. In both materials a clear negative scaling of the diffusivity with temperature is observed ($\partial D/\partial T < 0$), characteristic of band-like transport.

## Discussion

The observed correlation between diffusivity and lattice softness in combination with band-like transport is in good qualitative agreement with the formation of large exciton–polarons. However, further studies will be needed to provide a more quantitative model that can explain the large differences in diffusivity between the various organic spacers. The correct theoretical description of exciton–phonon coupling and exciton–polarons in 2D perovskites is still the subject of ongoing debate, though the current consensus is that the polar anharmonic lattice of these materials requires a description beyond conventional Frohlich theory[57,58,60]. Crucial in this respect will be further spectroscopic investigations of temperature-dependent optical properties of these materials, which should allow to better distinguish the influence of exciton–polaron formation from more traditional phonon-scattering mechanisms in these materials.

Meanwhile, structural rigidity can be used as a design parameter in these systems for optimized exciton transport characteristics. Taking into account the close correlation between diffusivity and the atomic displacement, this parameter space can be readily explored using available X-ray crystal structure data for many 2D perovskite analogues. While the influence of the organic spacer is expected to be particularly strong in the class of $n = 1$ 2D perovskites, we have observed consistent trends in the $n = 2$ analogues. Indeed, just like in $n = 1$, in $n = 2$ the use of the PEA cation yields higher diffusivities than for BA (see Supplementary Fig. 13). Similarly, the interstitial formamidinium (FA) cation in $n = 2$ yields higher diffusivity than the methylammonium (MA) cation, consistent with the trend in the atomic displacement parameters. It is important to note, though, that already for $n = 2$ perovskites a significant free carrier fraction may be present in the perovskites[61], suggesting that transport in $n > 1$ perovskites cannot be assumed to be purely excitonic and needs to be evaluated more rigorously.

From a technological perspective, structural rigidity may play a particularly important role in light emitting devices. Long exciton diffusion lengths in light emitting applications can act detrimentally on device performance, as it increases the possibility of encountering a trapping site. From an exciton–polaron perspective, this suggests soft lattices are preferred. At the same time though, Gong et al. highlighted the role of structural rigidity in improving the luminescence quantum yield through a reduced coupling to non-radiative decay pathways[55,62]. A trade-off therefore exists in choosing the optimal rigidity for bright emission. Meanwhile, for light harvesting applications, long diffusion lengths are essential for the successful extraction of excitons. While strongly excitonic 2D perovskites are generally to be avoided due to the penalty imposed by the exciton binding energy, improving the understanding of the spatial dynamics of the excitonic state may help mitigate this negative impact of the thinnest members of the 2D perovskites in solar harvesting.

In summary, we have studied the spatial and temporal exciton dynamics in 2D metal-halide perovskites of the form $L_2PbI_4$. We show that excitons undergo an initial fast diffusion through the crystalline plane, followed by a slower subdiffusive regime as excitons get trapped. Traps can be efficiently filled through a continuous wave background illumination, extending the initial regime where excitons undergo normal diffusion. By varying the organic spacer L we find that the intrinsic diffusivity depends

sensitively on the stiffness of the lattice, revealing a clear correlation between the lattice rigidity and the diffusivity. Our results indicate that exciton–phonon interactions dominate the spatial dynamics of excitons in 2D perovskites. Moreover, the observations are consistent with the formation of large exciton–polarons.

During the review process we became aware of a related manuscript by Deng et al.[63] using transient-absorption microscopy to study excited-state transport in 2D perovskites, with a focus on the differences in the spatial dynamics as a function of layer thickness ($n = 1$ to 5).

## Methods

**Growth of single-crystalline flakes.** Chemicals were purchased from commercial suppliers and used as received (see Supplementary Methods). Layered perovskites, with the exception of $(HA)_2PbI_4$ and $(DA)_2PbI_4$, were synthesized under ambient laboratory conditions following the over-saturation techniques[43,44]. In a nutshell, the precursor salts LI, $PbI_2$, and AI were mixed in a stoichiometric ratio (2:1:0 for $n = 1$ and 2:2:1 for $n = 2$) and dissolved in γ-butyrolactone. The solution was heated to 70 °C and more γ-butyrolactone was added (while stirring) until all the precursors were completely dissolved. The resulting solutions were heated to 70 °C and the solvent was left to evaporate. After 2–3 days, millimeter sized crystals formed in the solution, which was subsequently cooled down to room temperature. For this study, we drop cast some of the remaining supersaturated solution on a glass slide, heated it up to 50 °C with a hotplate and after the solvent was evaporated, crystals with crystal sizes of up to several hundred microns were formed. The saturated solution can be stored and re-used to produce freshly grown 2D perovskites within several minutes. We would like to note that drop cast $n = 2$ solutions form several crystals with different $n$ values. However, $n = 2$ crystals can be easily isolated during the exfoliation (see next section) and the formation of $n = 2$ can be favored by preheating the substrate to 50 °C before drop casting.

$(HA)_2PbI_4$ and $(DA)_2PbI_4$ were synthesized by dissolving $PbI_2$ (100 mg) in HI (800 µl) through heavy stirring and heating the solution to 90 °C. After $PbI_2$ was completely dissolved a stoichiometric amount of the amine was added dropwise to the solution.

**Exfoliation.** The perovskite crystals of the thin film were mechanically exfoliated using the Scotch tape method (Nitto SPV 224). The exfoliation guarantees a freshly cleaved and atomically flat surface area for inspection, which is crucial to avoid emission from edge states and guarantee direct contact with the glass substrate. After several exfoliation steps, the crystals were transferred on a glass slide and were subsequently studied through the glass slide with a ×100 oil immersion objective (Nikon CFI Plan Fluor, NA = 1.3). A big advantage of this technique is that the perovskites are encapsulated through the glass slide from one side and by the bulk of the crystal from the other side. It is important to use thick crystals to guarantee good self-encapsulation and prevent premature degradation of the perovskite flakes to affect the measurement[43].

**X-ray diffraction.** X-ray diffraction (XRD) was performed with a PANanaltical X'Pert PRO operating at 45 kV and 40 mA using a copper radiation source ($\lambda = 1.5406$ Å). The polycrystalline perovskite films were prepared by drop casting the saturated perovskite solutions on a silicon zero diffraction plate.

**Temperature-dependent photoluminescence measurements.** Perovskite flakes were excited with a 385 nm light emitting diode (Thorlabs) and the emission spectrum was measured using a spectrograph and an EMCCD camera coupled to a spectrograph (Princeton Instruments, SpectraPro HRS-300, ProEM HS 1024BX3). Temperature of the flakes was varied with a Peltier element (Adaptive Thermal Management, ET-127-10-13-H1), using a PID temperature controller (Dwyer Instruments, Series 16C-3) connected to a type K thermocouple (Labfacility, Z2-K-1M) for feedback control and a fan for cooling.

**Lifetime measurements.** Perovskite flakes were excited with a 405 nm laser (PicoQuant LDH-D-C-405, PDL 800-D), which was focused down to a near-diffraction limited spot. The photoluminescence was collected with an APD (Micro Photon Devices PDM, 20 × 20 µm detector size). The laser and APD were synchronized using a timing board for time correlated single photon counting (PicoHarp 300).

**Diffusion measurements.** Exciton diffusion measurements were measured following the same procedure as Akselrod et al.[42,45]. In short, a near diffraction limited exciton population was created using a 405 nm laser (PicoQuant LDH-D-C-405, PDL 800-D) and a ×100 oil immersion objective (Nikon CFI Plan Fluor, NA = 1.3). Fluorescence of the exciton population was then imaged with a total ×330 magnification onto an avalanche photodiode (APD, Micro Photon Devices PDM) with a detector size of 20 µm. The laser and APD were synchronized using a timing board for time correlated single photon counting (Pico-Harp 300). The APD was capturing

an effective area of around $60 \times 60$ nm ($= 20\,\mu m/330$). The APD was scanned through the middle of the exciton population in 60 or 120 nm steps, recording a time trace in every point. To minimize the degradation of the perovskites through laser irradiation, the perovskite flakes were scanned using an x-y-piezo stage (MCL Nano-BIOS 100), covering an area of $5 \times 5\,\mu m$. Diffusion measurements were performed with a 40 MHz laser repetition rate and a laser fluence of 50 nJ cm$^{-2}$ unless stated otherwise. The time binning of the measurement was set to 4 ps before software binning was applied. For the temperature-dependent measurements, the temperature was varied with a silicon heater mat (RS PRO, 245-499), using a PID temperature controller (Dwyer Instruments, Series 16C-3) connected to a type K thermocouple (Labfacility, Z2-K-1M) for feedback control. Here, a silicon heater mat was chosen over the Peltier element as a Peltier element expands during the heating process and causes mechanical vibrations that lead to drift.

**Brownian motion simulations**. We have performed Brownian dynamics simulations of a single exciton diffusing in a field of traps, representing ideal (non-interacting) excitons in the dilute limit carried out in experiments. In these simulations, an exciton diffuses freely until it finds a trap, where it just stops. Free diffusion is modelled using the standard stochastic differential equation for Brownian motion in the Itô interpretation. If $\mathbf{r}(t)$ is the position of the exciton in the plane at time $t$, its displacement $\Delta\mathbf{r}$ over a time $\Delta t$ is given by,

$$\Delta\mathbf{r} = \sqrt{2D}d\mathbf{W}, \tag{3}$$

where $D$ is the free-diffusion coefficient and $d\mathbf{W}$ is taken from a Wiener process, such that $\langle d\mathbf{W}d\mathbf{W}\rangle = \Delta t$. Traps were scattered throughout the plane following a uniform random distribution. The exciton is considered to be trapped as soon its location gets closer than $R_{trap} = 1.2$ nm to the trap center. The value was taken from estimations of the exciton Bohr radius and corresponds to a trap area of 1.44 nm$^2$ [37]. In any case, in the dilute regime, the diffusion is not sensitive to the trap size $R_{trap}$, because the trap radius is much smaller than the average separation between traps, $R_{trap} \ll \lambda$. To numerically integrate the equation of motion, we used a simple second-order-accurate modification of the well-known Euler Maruyama algorithm: the BAOAB-Limit method[64]. Trajectories were computed for many independent excitons and the data was averaged to determine the MSD as a function of time. While the simulation of the MSD was done in two dimensions, we used the MSD in one dimension to match the experimental conditions:

$$MSD(t) = \frac{1}{2}\left(MSD_x(t) + MSD_y(t)\right).$$

## Data availability

The data supporting the findings of this study are available within the article and its Supplementary information. Extra data are available upon reasonable request to the corresponding author.

## Code availability

Correspondence and requests for codes used in the paper should be addressed to the corresponding author.

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

## Acknowledgements

This work has been supported by the Spanish Ministry of Economy and Competitiveness through The "María de Maeztu" Program for Units of Excellence in R&D (MDM-2014-0377). M.S. acknowledges the financial support of a fellowship from "la Caixa" Foundation (ID 100010434). The fellowship code is LCF/BQ/IN17/11620040. M.S. has received funding from the European Union's Horizon 2020 research and innovation program under the Marie Skłodowska-Curie grant agreement No. 713673. F.P. acknowledges support from the Spanish Ministry for Science, Innovation, and Universities through the state program (PGC2018-097236-A-I00) and through the Ramón y Cajal program (RYC-2017-23253), as well as the Comunidad de Madrid Talent Program for Experienced Researchers (2016-T1/IND-1209). N.A., M.M. and R. D.-B. acknowledges support from the Spanish Ministry of Economy, Industry and Competitiveness through Grant FIS2017-86007-C3-1-P (AEI/FEDER, EU). E.P. acknowledges support from the Spanish Ministry of Economy, Industry and Competitiveness through Grant FIS2016-80434-P (AEI/FEDER, EU), the Ramón y Cajal program (RYC-2011- 09345) and the Comunidad de Madrid through Grant S2018/NMT-4511 (NMAT2D-CM). S.P. acknowledges financial support by the VILLUM FONDEN via the Centre of Excellence for Dirac Materials (Grant No. 11744).

## Author contributions

M.S. and F.P. designed this study. M.S. led the experimental work and processing of experimental data. M.S. set up the diffusion measurement technique with the assistance of T.J.L., and S.W.W. A.J.M. and M.S. performed temperature-dependent measurements. M.S. and A.J.M. prepared perovskite materials. N.A., M.M., and R.D.-B. performed theoretical and numerical modelling of exciton transport. M.S, F.P., S.P., and E.P. provided the theoretical interpretation of the intrinsic exciton transport. F.P. supervised the project. M.S. and F.P. wrote the original draft of the paper. All authors contributed to reviewing the paper.

## Competing Interests

The authors declare no competing interests.
