## [Peer Review File · Nature Communications]

Reviewers' comments:

Reviewer #1 (Remarks to the Author):

Seitz et al. use transient photoluminescence microscopy to probe exciton diffusion in 2D metal halide perovskites. The authors find a correlation between exciton diffusion length and lattice rigidity, which is altered by employing organic spacer ligands. By varying the background excitation intensity, the authors show that trap state filling can fully eliminate subdiffusion of excitons. To my knowledge, this is the first study to use a time-resolved optical microscopy technique to image exciton diffusion in 2D perovskites.

The manuscript is well written and thorough, especially considering the scope encompasses 2D perovskites with six different ligand spacers and also explores the dynamics of $n = 2$ perovskites. The technical execution and the analysis of the results are fine. I have a few questions regarding the relationship between the trap state density and the early time dynamics detailed below, but these are only for clarification rather than challenging the results and interpretations presented by the authors.

While exciton diffusion has been previously probed using indirect methods (fluence-dependent annihilation or quenching measurements), using transient photoluminescence microscopy to probe exciton diffusion directly is a timely and natural addition to the literature. This work will further contribute to understanding the importance of electron-phonon coupling on excitonic dynamics in 2D perovskites, and so after minor revisions, I can recommend it for publication in Nature Communications.

Questions:

- The authors demonstrate a clear correlation between average atomic displacement and the diffusion coefficient, implying that diffusivity is mostly controlled by the electron-phonon coupling and polaron formation. But ligands with different degrees of rigidity should also give rise to different trap state densities; for example, the quantum yield of phenethylammonium 2D perovskite thin films is higher than butylammonium. Could it now be possible that the quality of the crystals used could affect what the authors describe as a fundamental property of the material? In other words, can the authors fully rule out the influence of defects and traps on the 2D exciton diffusion, and why the early diffusion dynamics would be unaffected by the trap density?
- Can the authors compare the intrinsic trap state density they measure ($22 \mu\text{m}^{-2}$) with any literature values?
- How does disorder, as opposed to trap state density, affect the diffusion dynamics? Is it enough to say that the longer time subdiffusion arises solely due to trap states, and not disorder, since the trap states are filled and the subdiffusion is eliminated under additional background excitation? The authors should provide more detail about the trap states giving rise to these subdiffusive dynamics. In quantum dot solids, this subdiffusion is thought to arise due to energetic disorder (e.g. from dot size polydispersity). Such energetic disorder can also arise in two-dimensional materials, where fluctuations on the order of 100 meV can arise (<https://www.nature.com/articles/s41565-019-0520-0>), resulting in levels which also could be filled under higher excitation densities.

Reviewer #2 (Remarks to the Author):

Seitz et al. present a comprehensive and insightful study of exciton transport in layered 2D hybrid perovskites. Using transient photoluminescence microscopy, they measure the spatiotemporal dynamics of excitons in a variety of 2D perovskite structures and under varying experimental conditions (temperature, steady-state illumination). The most valuable new scientific insight is the role of the organic cation in electron-phonon scattering and its effect on exciton diffusivity, summarized in Fig. 3e and Fig. 4. Overall, the paper substantially extends our understanding of exciton photophysics in 2D perovskites and deserves to be published in Nature Communications.

It is worth noting the topical similarity to another study recently published in Nature Communications (Deng et al., Long-range exciton transport and slow annihilation in two-dimensional hybrid perovskites, Nature Commun (2020), <https://doi.org/10.1038/s41467-020-14403-z>). However, I find that there is very little overlap in these two papers and that they complement each other nicely. The study by Deng et al. focused on layer thickness, whereas the manuscript by Seitz et al focuses on organic cation composition, trap states, and temperature. Of the two, I find the present manuscript by Seitz et al to be more insightful. Deng et al. provided a very nice benchmark data set for subsequent comparison, but the work by Seitz et al provides more explanation of the physical origin of the observed behavior. Both papers deserve to be published side-by-side in Nature Communications.

A couple points of discussion:

1) It may be worthwhile for the authors to discuss the possible role of free carriers vs excitons in their observed transport dynamics. Following the study by Gelvez-Rueda et al (<https://doi.org/10.1021/acs.jpcc.7b10705>), a surprisingly large fraction of excited states in 2D perovskites may exist as free charges – rather than excitons. This is more likely to impact the interpretation of measured diffusivity in thicker 2D perovskites (such as those studied by Deng et al), but it may be a point worth mentioning in this work.

2) The discussion of polarons could be improved. While I personally agree with authors' perspectives, the subject and claims in the paper are possibly more controversial. The section spanning pages 12-13 could be improved if the authors first present the data in Figs. 3 & 4, then save their polaron interpretation for later in the Discussion section. As it's written now, the paper reads as if the experimental data are confirming polaronic behavior that was previously expected; it undersells the novelty of the experimental results and unnecessarily raises doubt about the interpretation before the actual data (which are not in doubt) are presented.

Reviewer #3 (Remarks to the Author):

In this manuscript entitled "Exciton diffusion in two-dimensional metal-halide perovskites", Michael et al. presented a comprehensive study of exciton diffusion in two dimensional perovskites by transient photoluminescence microscopy. Exciton diffusion was shown to be strongly dependent on the organic ligands. Subdiffusive behavior of exciton at long-delay time was attributed to the trapping by defects. The manuscript is well written with high-quality data and the conclusions are for the most part supported by the experiments.

I have the following suggestions for the authors to consider:

1. It is remarkable that the exciton diffusion constant with PEA ligands is one order of magnitude higher than that with the BA ligand. However, it is unlikely that the polaron mass (which is determined

by the electron-phonon coupling strength) is 10 times heavier in BA than PEA. Therefore, it is likely that other factors also play a role. For example, the optical phonon modes in BA probably are lower in frequency (softer) – will that play a role? Also, a softer lattice could also mean large structural disorder. Some further discussion on this point will be helpful.

2. The temperature dependence data shown in Fig.4 is of a very narrow range and the error bar for the PEA sample is missing. So I am not sure if conclusions regarding coherent/ incoherent transport can be convincingly drawn.

3. In fig.1d, it seems that the fittings of curves with large delay time give non-negligible positive value backgrounds (y_0) in addition to the peaks. I suppose the curves should have 0 backgrounds otherwise the sigmas could be underestimated. Detail of fitting should be provided.

4. The authors should compare their results with previous reported carrier diffusivities (mobilities) in two dimensional perovskites.

5. Some typos: In figure 3b, $MSD(t)^2$ should be $MSD(t)$; in line 185, it should be fig.3b; in line 187, it should be fig.3c.

Reviewer #1 (Remarks to the Author):

Seitz et al. use transient photoluminescence microscopy to probe exciton diffusion in 2D metal halide perovskites. The authors find a correlation between exciton diffusion length and lattice rigidity, which is altered by employing organic spacer ligands. By varying the background excitation intensity, the authors show that trap state filling can fully eliminate subdiffusion of excitons. To my knowledge, this is the first study to use a time-resolved optical microscopy technique to image exciton diffusion in 2D perovskites.

The manuscript is well written and thorough, especially considering the scope encompasses 2D perovskites with six different ligand spacers and also explores the dynamics of $n = 2$ perovskites. The technical execution and the analysis of the results are fine. I have a few questions regarding the relationship between the trap state density and the early time dynamics detailed below, but these are only for clarification rather than challenging the results and interpretations presented by the authors.

While exciton diffusion has been previously probed using indirect methods (fluence-dependent annihilation or quenching measurements), using transient photoluminescence microscopy to probe exciton diffusion directly is a timely and natural addition to the literature. This work will further contribute to understanding the importance of electron-phonon coupling on excitonic dynamics in 2D perovskites, and so after minor revisions, I can recommend it for publication in Nature Communications.

We thank the reviewer for taking the time and effort to provide this feedback and we appreciate the positive evaluation. We will address the reviewer's specific points on the following pages.

Questions:

- The authors demonstrate a clear correlation between average atomic displacement and the diffusion coefficient, implying that diffusivity is mostly controlled by the electron-phonon coupling and polaron formation. But ligands with different degrees of rigidity should also give rise to different trap state densities; for example, the quantum yield of phenethylammonium 2D perovskite thin films is higher than butylammonium. Could it now be possible that the quality of the crystals used could affect what the authors describe as a fundamental property of the material? In other words, can the authors fully rule out the influence of defects and traps on the 2D exciton diffusion, and why the early diffusion dynamics would be unaffected by the trap density?

We thank the reviewer for raising this point since it is a central part of our study. Two observations lead us to the conclusion that the early time dynamics is unaffected by the trap state density. First, the early time dynamics show a linear behavior between the mean-square-displacement (MSD) and time. Linear diffusion is a clear indication of unobstructed diffusion, since any trap density would lead to sublinear behavior. Second, and most importantly, in Figure 2 of the main text we directly investigate the influence of trap states on exciton diffusion by actively changing the effective number of traps in the perovskite. We find that reducing the trap density only affects the late time dynamics and has no observable influence on the early time dynamics.

We have added the following sentences to line 182 in order to highlight this point:

"Importantly, the early diffusion dynamics is unaffected by the trap density and shows normal diffusion ($\alpha = 1$) for all illumination intensities. This strongly suggests that the early diffusion dynamics is unaffected by energetic disorder, which would result in a sublinear behavior with $\alpha < 1$,

and any trap states, giving us direct access to the intrinsic exciton diffusivity of the material and allowing us to compare the intrinsic exciton diffusivity between perovskites of different compositions.”

For completeness we have included a full analysis of the trap density of (BA)₂PbI₄ to the SI as a new Figure S3. We find the trap density in (BA)₂PbI₄ to be higher than in (PEA)₂PbI₄ (204 μm⁻² vs. 22 μm⁻²). Nevertheless, a clear linear ($\alpha = 1$) behavior at early times can also be identified for (BA)₂PbI₄.

- Can the authors compare the intrinsic trap state density they measure (22 μm⁻²) with any literature values?

We thank the reviewer for this suggestion and agree that a comparison to literature values will strengthen the manuscript. In general, reported values for trap state densities in perovskites vary greatly with the characterization method. However, our reported trap density of 22 μm⁻² per layer ($\approx 10^{16}$ cm⁻³) is around the same order as previously reported values for bulk perovskites measured with optical methods. We added the following sentence to the manuscript in lines 161-163:

“... which, as shown in Fig. 2c, successfully reproduces both experimental and numerical results and allows us to determine the value of the intrinsic trap state density, yielding $1/\lambda^2 = 22 \mu\text{m}^{-2}$ **per layer ($\approx 10^{16} \text{cm}^{-3}$)**, which is of the same order of magnitude as previously reported values for bulk perovskites.^{48,49}”

- How does disorder, as opposed to trap state density, affect the diffusion dynamics? Is it enough to say that the longer time subdiffusion arises solely due to trap states, and not disorder, since the trap states are filled and the subdiffusion is eliminated under additional background excitation? The authors should provide more detail about the trap states giving rise to these subdiffusive dynamics. In quantum dot solids, this subdiffusion is thought to arise due to energetic disorder (e.g. from dot size polydispersity). Such energetic disorder can also arise in two-dimensional materials, where fluctuations on the order of 100 meV can arise (<https://www.nature.com/articles/s41565-019-0520-0>), resulting in levels which also could be filled under higher excitation densities.

We thank the reviewer for this question. Indeed, energetic disorder can play an important role in exciton diffusion. As the reviewer points out, our observation of a full elimination of subdiffusive behaviour in the trap-state filling experiment is indeed an important indication that traps are the dominant factor. Energetic disorder, as is present in for example quantum dot solids with size-polydispersity, would lead to an overall sub-diffusive behavior (as observed in for example ref. 45 of the main text). Our observation of an early regime of normal diffusion, followed by a subdiffusive regime is indicative of trap-state limited diffusion (as observed in for example ref. 42 of the main text for organic single crystals). The delay in the onset of subdiffusion reflects the average time it takes for excitons to meet a trap – similar to a mean free path. It should be stressed that this is corroborated by the good agreement between the experiments and the theoretical model, as well as Brownian dynamics simulations.

Reviewer #2 (Remarks to the Author):

Seitz et al. present a comprehensive and insightful study of exciton transport in layered 2D hybrid perovskites. Using transient photoluminescence microscopy, they measure the spatiotemporal dynamics of excitons in a variety of 2D perovskite structures and under varying experimental conditions (temperature, steady-state illumination). The most valuable new scientific insight is the role of the organic cation in electron-phonon scattering and its effect on exciton diffusivity, summarized in Fig. 3e and Fig. 4. Overall, the paper substantially extends our understanding of exciton photophysics in 2D perovskites and deserves to be published in Nature Communications.

It is worth noting the topical similarity to another study recently published in Nature Communications (Deng et al., Long-range exciton transport and slow annihilation in two-dimensional hybrid perovskites, Nature Commun (2020), <https://doi.org/10.1038/s41467-020-14403-z>). However, I find that there is very little overlap in these two papers and that they complement each other nicely. The study by Deng et al. focused on layer thickness, whereas the manuscript by Seitz et al focuses on organic cation composition, trap states, and temperature. Of the two, I find the present manuscript by Seitz et al to be more insightful. Deng et al. provided a very nice benchmark data set for subsequent comparison, but the work by Seitz et al provides more explanation of the physical origin of the observed behavior. Both papers deserve to be published side-by-side in Nature Communications.

We thank the reviewer for taking the time and effort to provide this feedback. We appreciate the positive assessment and the valuable suggestions. We will address the reviewer's specific points on the following pages.

A couple points of discussion:

1) It may be worthwhile for the authors to discuss the possible role of free carriers vs excitons in their observed transport dynamics. Following the study by Gelvez-Rueda et al (<https://doi.org/10.1021/acs.jpcc.7b10705>), a surprisingly large fraction of excited states in 2D perovskites may exist as free charges – rather than excitons. This is more likely to impact the interpretation of measured diffusivity in thicker 2D perovskites (such as those studied by Deng et al), but it may be a point worth mentioning in this work.

We fully agree with the reviewer. Indeed, Gelvez-Rueda et al. showed that assuming a purely excitonic nature of 2D perovskites is only applicable for the thinnest $n = 1$ perovskites. For $n = 2$ the free carrier fraction is already $\sim 50\%$ and keeps increasing with increasing thickness ($n > 2$). We have added the following comment to the main text to highlight this point in lines 281 – 283:

"It is important to note, though, that already for $n = 2$ perovskites a significant free carrier fraction may be present in the perovskites,⁶³ suggesting that transport in $n > 1$ perovskites cannot be assumed to be purely excitonic and needs to be evaluated more rigorously."

In addition, we have added a note on the study by Deng et al at the end of the discussion (lines 295-297):

“During the review process we became aware of a related manuscript by Deng et al.⁶⁵ using transient-absorption microscopy to study excited-state transport in 2D perovskites, with a focus on the differences in the spatial dynamics as a function of layer thickness ($n = 1-5$).”

2) The discussion of polarons could be improved. While I personally agree with authors' perspectives, the subject and claims in the paper are possibly more controversial. The section spanning pages 12-13 could be improved if the authors first present the data in Figs. 3 & 4, then save their polaron interpretation for later in the Discussion section. As it's written now, the paper reads as if the experimental data are confirming polaronic behavior that was previously expected; it undersells the novelty of the experimental results and unnecessarily raises doubt about the interpretation before the actual data (which are not in doubt) are presented.

We thank the reviewer for this suggestion and agree that the formation of exciton-polarons in 2D perovskites is still subject for debate. Following the suggestion of the reviewer, we have restructured the text. Most importantly, the original paragraph introducing the polaron formation has been moved further down and now more explicitly mentions the debate on the existence of exciton-polarons in 2D perovskites. Moreover, we have added a paragraph to the discussion to present the polaron interpretation in more detail.

The paragraph introducing figure 3 now starts in line 223 with:

“Recently, exciton-phonon interactions have been found to strongly influence exciton dynamics in perovskites.^{31,33} To investigate the possible role of exciton-phonon coupling on exciton diffusion, (...)”

The introduction to polaron formation has been moved to the section concerning figure 4, which now reads (lines 242-256):

“In the limit of strong exciton-phonon coupling, the presence of an exciton could potentially cause distortions of the soft inorganic lattice of the perovskite and lead to the formation of exciton-polarons.^{59,60} As compared to a free exciton, an exciton-polaron would exhibit a larger effective mass and, consequently, a lower diffusivity. The softer the lattice, the larger the distortion, and the heavier the polaron effective mass would be.⁶¹”

Polaron formation can significantly modify the mechanism of transport, in some cases causing a transition from band-like to a hopping type transport.⁶¹ When short-range deformations of the lattice are dominant, the exciton-polaron is localized within a unit cell of the material and is known as a *small* polaron. The motion of *small* polarons occurs through site-to-site hopping and increases with temperature ($\partial D/\partial T > 0$). However, in the presence of dominant long-range lattice deformations, *large* exciton-polarons may form which extend across multiple lattice sites. The diffusion of *large* polarons decreases with increasing temperature ($\partial D/\partial T < 0$), resembling that of band-like free exciton motion, although with a strongly increased effective mass. In Fig. 4, we present temperature dependent measurements of the diffusivity for both $(\text{PEA})_2\text{PbI}_4$ and $(\text{BA})_2\text{PbI}_4$. In both materials a clear negative scaling of the diffusivity with temperature is observed ($\partial D/\partial T < 0$), characteristic of band-like transport.”

A new paragraph has been introduced in the discussion and reads (lines 263-272):

“The observed correlation between diffusivity and lattice softness in combination with band-like transport is in good qualitative agreement with the formation of large exciton-polarons. However, further studies will be needed to provide a more quantitative model that can explain the large differences in diffusivity between the various organic spacers. The correct theoretical description of exciton-phonon coupling and exciton-polarons in 2D perovskites is still the subject of ongoing debate, though the current consensus is that the polar anharmonic lattice of these materials requires a description beyond conventional Frohlich theory.^{59,60,62} Crucial in this respect will be further spectroscopic investigations of temperature dependent optical properties of these materials, which should allow to better distinguish the influence of exciton-polaron formation from more traditional phonon-scattering mechanisms in these materials.”

Reviewer #3 (Remarks to the Author):

In this manuscript entitled “Exciton diffusion in two-dimensional metal-halide perovskites”, Michael et al. presented a comprehensive study of exciton diffusion in two dimensional perovskites by transient photoluminescence microscopy. Exciton diffusion was shown to be strongly dependent on the organic ligands. Subdiffusive behavior of exciton at long-delay time was attributed to the trapping by defects.

The manuscript is well written with high-quality data and the conclusions are for the most part supported by the experiments.

We thank the reviewer for taking the time and effort to provide this feedback. We appreciate the thoroughness of this review.

I have the following suggestions for the authors to consider:

1. It is remarkable that the exciton diffusion constant with PEA ligands is one order of magnitude higher than that with the BA ligand. However, it is unlikely that the polaron mass (which is determined by the electron-phonon coupling strength) is 10 times heavier in BA than PEA. Therefore, it is likely that other factors also play a role. For example, the optical phonon modes in BA probably are lower in frequency (softer) – will that play a role? Also, a softer lattice could also mean large structural disorder. Some further discussion on this point will be helpful.

We thank the reviewer for this comment. We agree that a quantitative evaluation of the influence of lattice rigidity on the diffusivity is needed (indeed regarding the details of the type of phonons and different types of scattering mechanisms). However, to answer these questions, an accurate theoretical description of the exciton-polaron is needed, a model for which is unfortunately lacking at the moment. To emphasize these aspects, we have added the following paragraph to the discussion in lines 263-272:

“The observed correlation between diffusivity and lattice softness in combination with band-like transport is in good qualitative agreement with the formation of large exciton-polarons. However, further studies will be needed to provide a more quantitative model that can explain the large differences in diffusivity between the various organic spacers. The correct theoretical description of exciton-phonon coupling and exciton-polarons in 2D perovskites is still the subject of ongoing debate, though the current consensus is that the polar anharmonic lattice of these materials requires a description beyond conventional Frohlich theory.^{59,60,62} Crucial in this respect will be further spectroscopic investigations of temperature dependent optical properties of these materials, which should allow to better distinguish the influence of exciton-polaron formation from more traditional phonon-scattering mechanisms in these materials.”

2. The temperature dependence data shown in Fig.4 is of a very narrow range and the error bar for the PEA sample is missing. So I am not sure if conclusions regarding coherent/ incoherent transport can be convincingly drawn.

We agree with the reviewer that a bigger temperature range is needed to draw more detailed conclusions about the mechanism of transport. Unfortunately, our current diffusion measurements are done using an oil immersion objective, preventing us from measuring across wider temperature ranges. Nevertheless, the significant decrease in diffusivity (logarithmic scale in Fig. 4) with increasing

temperature shows a clear negative temperature dependence ($\partial D/\partial T < 0$), characteristic of band-like transport. Following the suggestion of the reviewer, we have rephrased the conclusions regarding the temperature dependence, removing any reference to the coherent and/or incoherent nature of the transport (lines 247-256):

“Polaron formation can significantly modify the mechanism of transport, in some cases causing a transition from band-like to a hopping type transport.⁶¹ When short-range deformations of the lattice are dominant, the exciton-polaron is localized within a unit cell of the material and is known as a *small* polaron. The motion of *small* polarons occurs through site-to-site hopping and increases with temperature ($\partial D/\partial T > 0$). However, in the presence of dominant long-range lattice deformations, *large* exciton-polarons may form which extend across multiple lattice sites. The diffusion of *large* polarons decreases with increasing temperature ($\partial D/\partial T < 0$), resembling that of band-like free exciton motion, although with a strongly increased effective mass. In Fig. 4, we present temperature dependent measurements of the diffusivity for both $(\text{PEA})_2\text{PbI}_4$ and $(\text{BA})_2\text{PbI}_4$. In both materials a clear negative scaling of the diffusivity with temperature is observed ($\partial D/\partial T < 0$), characteristic of band-like transport.”

In Fig. 4 the error bars for PEA are smaller than the marker, therefore they are not visible in the plot. We have added a comment to the caption of Fig. 4 to emphasize this:

“Error bars represent the uncertainty of the fit and are smaller than the markers for $(\text{PEA})_2\text{PbI}_4$.”

3. In fig.1d, it seems that the fittings of curves with large delay time give non-negligible positive value backgrounds (γ_0) in addition to the peaks. I suppose the curves should have 0 backgrounds otherwise the sigmas could be underestimated. Detail of fitting should be provided.

We thank the reviewer for this comment. We agree that the fitting procedure requires careful consideration of the offset due to background noise. In our fitting procedure we do this by using a global offset (γ_0), which is the same for all times t . The reason why the offset appears to be bigger for $t = 10\text{ns}$, than for $t = 0\text{ ns}$ is that the data and fitted curves are normalized to 1 in Fig. 1d. However, while we normalized the curves in Fig. 1d to highlight the broadening of the exciton distribution, we use the unnormalized data in our fitting procedure to correctly account for the background noise. We describe the fitting procedure in detail in the SI in the “2. Exciton Diffusion Measurements” section (page 7).

For clarification we added the following sentence to page 8 of the SI:

“Fig. 1d shows a subset of the fitted Voigt functions together with the experimental data, which were normalized to 1 for a better comparison. Please note that the real offset for all curves is the same. The apparent increase of the offset for later times is solely due to the normalization of the data.”

4. The authors should compare their results with previous reported carrier diffusivities (mobilities) in two dimensional perovskites.

We thank the reviewer for this suggestion. We added the following sentence to the main text in lines 131-136:

“Our diffusivity of single crystalline $(\text{PEA})_2\text{PbI}_4$ is around an order of magnitude higher than previously reported mobility values from conductivity measurements ($\mu = 1\text{ cm}^2/\text{V/s}$; $D = \mu k_B T = 0.025\text{ cm}^2/\text{s}$) of polycrystalline films.⁴⁶ This finding is reasonable as grain boundaries slow down the movement of excitons and conventional methods measure a time-averaged mobility that cannot separate intrinsic

diffusion from trap-state limited diffusion.”

5. Some typos: In figure 3b, $MSD(t)^2$ should be $MSD(t)$; in line 185, it should be fig.3b; in line 187, it should be fig.3c.

We thank the reviewer for the careful reading of our manuscript. We corrected the typos as suggested by the reviewer.

REVIEWERS' COMMENTS:

Reviewer #1 (Remarks to the Author):

The authors have acted on referee suggestions. The work is appropriate for Nature Comms.

Reviewer #2 (Remarks to the Author):

I thank the authors for carefully considering the reviewers' suggestions and improving their paper accordingly. I think the re-written section on exciton-polarons is much stronger now and more accurately reflects consensus thinking in the community.

I would be pleased to see this work published in Nature Communications.

Reviewer #3 (Remarks to the Author):

The revision has satisfactorily addressed my comments as well as other reviewers' comments.